# OpenReview forum: "OpenMathInstruct-2: Accelerating AI for Math with Massive Open-Source Instruction Data"
_ICLR.cc/2025/Conference — ICLR 2025 Poster_

### Official Review · Reviewer_uMxv · 2024-11-01

**Soundness:** 3
**Presentation:** 3
**Contribution:** 3
**Rating:** 6
**Confidence:** 3

**Summary:**

This paper curate a large scale (14 M qa-pairs) and high-quality math reasoning SFT datasets - OpenMathInstruct-2. At the same time, authors find some insights:

Verbose solutions is detrimental to SFT performance. Data generated by strong model is better. SFT allow some low-quality solutions. Question diveristy is crucial.

SFT the Llama-3.1-8B-Base on OpenMathInstruct-2 dataset achieve 67.8% performance, outperforming Llama3.1-8B-Instuct by 15.9%

**Strengths:**

The paper introduces a large-scale math reasoning dataset, which would be highly beneficial if released.

It is well-written and easy to follow.

The authors offer valuable insights into data curation and supervised fine-tuning, discussing aspects like the format of chain-of-thought and the diversity of questions. Some insights are interesting and can be explored deeper.

**Weaknesses:**

The process of dataset generation is not novel. The main advantage is the larger scale.

Some findings like data generated by strong teacher is better, question diversity matters are very natural.

**Questions:**

How do authors find or select the OpenMath CoT format? The findings that OpenMath CoT is short but performs better is interesting. I think better COT format exploration is an interesting direction.

In section 3.1, aurthors discussed the LLM decontamination. Is the SFT data is processed the decontamnation? Have authors conducted experiments comparing the performance of with and without the decontamination?

---

> ### Author Response · Authors · 2024-11-21
> **Response to Reviewer uMxv**
>
> We thank the reviewer for their feedback and have answered their questions below.
>
> > The paper introduces a large-scale math reasoning dataset, which would be highly beneficial if released.
>
> We are committed to releasing the dataset, code, and finetuned models.
>
> > Some findings like data generated by strong teacher is better, question diversity matters are very natural.
>
> While these findings can seem obvious, prior literature lacks the controlled setup that we present. Our ablation concerning the choice of teacher model shows that even when controlling for the quantity of the SFT data, i.e. matching the number of solutions for all the questions, the quality of solutions generated by a strong teacher model can still significantly improve the finetuning performance. Moreover, we also present findings on the importance of CoT format and robustness of SFT to noisy data which to the best of our knowledge are novel findings.
>
> > How do authors find or select the OpenMath CoT format? The findings that OpenMath CoT is short but performs better is interesting. I think better COT format exploration is an interesting direction.
>
> The OpenMath CoT format is something we refined over time based on our experiments. It’s essentially based on the few-shot prompts described in Section D.1. The solutions are broken down into step-by-step computation, but avoid the verbosity of Llama models such as starting each step with the prefix “Step N:”. The OpenMath CoT format also emphasizes decomposing complex arithmetic computations into simpler steps.
>
>
> > In section 3.1, aurthors discussed the LLM decontamination. Is the SFT data is processed the decontamnation? Have authors conducted experiments comparing the performance of with and without the decontamination?
>
> We find that without decontamination the SFT performance improves by about 1% for MATH which we attribute to test set contamination.

---

> > ### Comment · Reviewer_uMxv · 2024-11-26
> >
> > Thanks for author's response. I will increase the score.

---

### Official Review · Reviewer_vtSB · 2024-11-03

**Soundness:** 3
**Presentation:** 3
**Contribution:** 3
**Rating:** 6
**Confidence:** 5

**Summary:**

The paper introduces OpenMathInstruct-2, a large open-source dataset for mathematical reasoning containing 14M question-solution pairs (approximately 600K unique questions), making it nearly eight times larger than previous open-source datasets. The authors conducted systematic experiments to understand how different dataset design choices affect model performance, finding that:

1. Solution format matters - excessive verbosity can hurt performance.
2. Teacher model quality is important
3. SFT is robust to low-quality solutions
4. Question diversity is crucial.

Using these insights, they created OpenMathInstruct-2 and demonstrated its effectiveness through model fine-tuning. Their OpenMath2-Llama3.1-8B model achieved 67.8% accuracy on MATH, outperforming Llama3.1-8B-Instruct by 15.9%. The authors also introduced a thorough LLM-based decontamination pipeline to ensure test set integrity. The paper concludes by releasing all code, models, and the dataset under a permissive license to support open-source development in mathematical reasoning.

**Strengths:**

* **Ablation Studies & Methodology** - The paper excels in its systematic investigation of dataset design choices. The ablation studies on solution format, teacher model quality, and robustness to low-quality solutions are thorough and well-controlled.

* **Open Science & Reproducibility** - The authors make significant contributions to open-source development by releasing their dataset, code, models, and detailed methodology. Their comprehensive documentation of prompts (in appendices D.1-D.5) and data generation pipelines enable full reproducibility.

* **Math Synthetic Data Insights** - The paper provides several interesting findings that advance our understanding of LLM synthetic data generation in math reasoning. For instance, the discovery that shorter solution formats can outperform more verbose ones (OpenMath CoT vs Llama CoT) challenges conventional wisdom about detailed explanations. Similarly, their finding that SFT is robust to up to 20% incorrect solutions provides valuable guidance for dataset creation, suggesting that aggressive filtering may be unnecessary.

**Weaknesses:**

**Insufficient Exploration of Question Generation Quality** - While the paper presents a thorough decontamination pipeline for synthetic questions, it lacks a detailed analysis of the quality and diversity distribution of generated questions. The authors don't evaluate whether their synthetic questions maintain: 1) the same quality as the original questions; 2) the same difficulty level as the original dataset 3) the same diversity (or more diverse) compared with the seed data. This might be crucial. The authors' ablation studies also show that even though the model is error-tolerant to wrong answers to some extent (e.g., 20%), the accuracy would downgrade more if there were too many error examples in the training. It would be necessary to examine the quality of the 14M generated samples.


**Potential Knowledge Distillation Bottleneck** - There's a fundamental question about the approach. Though the performance is promising on a small student model (i.e., Llama-3.1-8B) with the 14M synthetic data, it is unclear whether improvement comes from primarily distilling the teacher's knowledge or improving genuine mathematical reasoning capabilities. The diminishing returns with larger student models (70B vs 8B) are consistent with distillation behavior, where the student model's capacity approaches needed to mimic the teacher's surface patterns rather than learning deeper reasoning. This limitation could be particularly important for scaling mathematical reasoning capabilities beyond the current teacher model's abilities. I'd suggest either of the experiments:
- Further fine-tune the teacher model (340B) on the 14M data (which might be costly)
- Generate a relatively larger-scale training set (>1M) using an 8B model and fine-tune the model on the generated data.

**Incomplete Investigation of Teacher Model Selection** - The authors demonstrate that Llama3.1-405B-Instruct outperforms Llama3.1-8B-Base as a teacher model, but they don't explore the space between these two extremes. Would a 70B model be sufficient for a teacher? Is there a clear relationship between teacher model size and generated solution quality? Given the computational cost implications of using a 405B parameter model for data generation, understanding the minimum viable teacher model size would be practically valuable.

**Questions:**

- The authors acknowledge but don't fully investigate why their OpenMath2-Llama3.1-70B model shows less dramatic improvements compared to their 8B model. They hypothesize that "our data blend or solution format might be more suited for weaker models" but don't test this hypothesis.

- Have you tried other sizes of teacher models?

---

> ### Author Response · Authors · 2024-11-21
> **Response to Reviewer vtSB - Part 1**
>
> We thank the reviewer for their detailed feedback and thoughtful follow-up questions.
>
> > Insufficient Exploration of Question Generation Quality - While the paper presents a thorough decontamination pipeline for synthetic questions, it lacks a detailed analysis of the quality and diversity distribution of generated questions. The authors don't evaluate whether their synthetic questions maintain: 1) the same quality as the original questions; 2) the same difficulty level as the original dataset 3) the same diversity (or more diverse) compared with the seed data. This might be crucial. The authors' ablation studies also show that even though the model is error-tolerant to wrong answers to some extent (e.g., 20%), the accuracy would downgrade more if there were too many error examples in the training. It would be necessary to examine the quality of the 14M generated samples.
>
> This is an intriguing question. To answer this, we compare the SFT performance on a subset created with the original training set using solution augmentation and create comparable subsets from synthesized question-solution pairs. So, OpenMathInstruct-2 has solutions for 7436/7500 training set problems in MATH. We create a 128K-sized, fair-downsampled subset for question-solution pairs created using these original training set problems (only the solutions are synthesized). We sample four random subsets of 7436 synthesized questions from OpenMathInstruct-2 and use their corresponding solutions from OpenMathInstruct-2 to create 128K-sized, fair-downsampled subsets. We finetune the Llama3.1-8B-Base on these five subsets and present the result in the table below where we refer to the subset with original questions as Original and the subsets with synthesized questions as Synthetic.
>
> | Question Source| MATH Test Accuracy |
> |-----------|-------------------|
> | Original  | 44.10            |
> | Synthetic | 39.79 ± 0.53     |
>
> We see that there’s a reasonable difference between the SFT performance suggesting a drop in the quality of synthetic questions from the original questions. We hypothesize this could be due to the following reasons: (a) creating variations of certain questions is much easier than others because of which we might have underrepresentation of certain question types among the synthetic questions, and (b) for synthetic questions we are treating majority answer as the ground truth answer which can be incorrect. Future work can look at look at improving the quality of synthetic questions-answers by employing powerful reward models such as Qwen-72B-Instruct to clean the data.
>
> While lower in quality than the original training set questions, we want to emphasize the importance of question-solution augmentation to the huge performance impact of OpenMathInstruct-2. With just solution augmentation, we reach a performance of about 51 on MATH with a SFT dataset of 2.5 million samples. By including the QA pairs from question-solution augmentation, we reach a performance of 61.2 with just 1 million samples.
>
> Regarding estimating the difficulty of the generated questions, while it’s possible to estimate the difficulty by evaluating the existing models on this data, the noisy ground truth answer makes this estimation unreliable.
>
> Finally, to calculate the diversity of the questions, we use the multi-qa-MiniLM-L6-cos-v1 to embed all the questions in the different subsets and calculate the average cosine similarity between the questions.
>
> | Question Source| Average Cosine Similarity |
> |-----------|-------------------|
> | Original  | 0.13            |
> | Synthetic | 0.12 ± 0.00   |
>
> As can be seen from the above table, the average cosine similarity of the two original and synthetic subsets is quite comparable. In fact, the synthetic subsets are marginally more diverse.

---

> > ### Author Response · Authors · 2024-11-21
> > **Response to Reviewer vtSB - Part 2**
> >
> > > Potential Knowledge Distillation Bottleneck - There's a fundamental question about the approach. Though the performance is promising on a small student model (i.e., Llama-3.1-8B) with the 14M synthetic data, it is unclear whether improvement comes from primarily distilling the teacher's knowledge or improving genuine mathematical reasoning capabilities. The diminishing returns with larger student models (70B vs 8B) are consistent with distillation behavior, where the student model's capacity approaches needed to mimic the teacher's surface patterns rather than learning deeper reasoning. This limitation could be particularly important for scaling mathematical reasoning capabilities beyond the current teacher model's abilities. I'd suggest either of the experiments:
> > >- Further fine-tune the teacher model (340B) on the 14M data (which might be costly)
> > >- Generate a relatively larger-scale training set (>1M) using an 8B model and fine-tune the model on the generated data.
> >
> > Since we’re training only on the solution trajectories that get to the (pseudo-)ground truth answers, the student model, in theory, can exceed the teacher model’s greedily sampled reasoning capabilities. As suggested by the reviewer and as has been demonstrated by plenty of prior work, we show that a Llama3.1-8B-Base model can perform self-improvement i.e. exceed the teacher’s performance. We finetune the Llama3.1-8B-Base on 860K synthetic solutions synthesized by the model itself which lifts the performance of the model from 20.3% to 30.5% on MATH. Unfortunately, finetuning the Llama 405B model is not possible with our current resources, so we limit the self-improvement experiment to Llama3.1-8B-Base.
> >
> > Regarding whether the training is "improving genuine mathematical reasoning capabilities," we believe such questions can only be answered via evaluations. Our current results suggest that results on benchmarks other than GSM8K and MATH also improve after finetuning with OpenMathInstruct-2. For example, after finetuning the Llama3.1-8B-Base model with OpenMathInstruct-2, the performance on AIME 2024 goes from 0/30 to 3/30.
> >
> > > Incomplete Investigation of Teacher Model Selection - The authors demonstrate that Llama3.1-405B-Instruct outperforms Llama3.1-8B-Base as a teacher model, but they don't explore the space between these two extremes. Would a 70B model be sufficient for a teacher? Is there a clear relationship between teacher model size and generated solution quality? Given the computational cost implications of using a 405B parameter model for data generation, understanding the minimum viable teacher model size would be practically valuable.
> >
> > We agree with the reviewer that there’s indeed a broad spectrum of model capabilities between Llama3.1-8B-Base and Llama3.1-405B-Instruct, and future work can explore in more depth the task of identifying the best teacher model/combination of teacher models. For now, we present results preliminary results based on 128K-sized, fair-downsampled SFT data created using solution augmentation via the Llama3.1-{8B/70B/405B}-Instruct models.
> >
> > | Teacher model | MATH Test Accuracy |
> > | ------------- | ------------------ |
> > | 8B-Instruct   | 42.96             |
> > | 70B-Instruct  | 44.40             |
> > | 405B-Instruct | 44.44             |
> >
> > The Llama3.1-70B-Instruct and Llama3.1-405B-Instruct models are quite comparable and marginally better than the Llama3.1-8B-Instruct model as teacher.
> >
> >
> > > The authors acknowledge but don't fully investigate why their OpenMath2-Llama3.1-70B model shows less dramatic improvements compared to their 8B model. They hypothesize that "our data blend or solution format might be more suited for weaker models" but don't test this hypothesis.
> >
> > Due to the cost of fully finetuning the 70B model, we are limited in our capability to conduct a more thorough investigation on suggested hyperparameters. Note that the 70B model is also trained on a 5M subset of the data which might partially explain the less dramatic improvements for 70B (still an increase of 4% on MATH over Llama3.1-70B). We leave such investigations focussed on bigger models for future work.

---

> > > ### Comment · Reviewer_vtSB · 2024-11-23
> > >
> > > We thank the authors for their efforts in answering my queries.
> > >
> > > - From the new results, It seems that 70B model achieves comparable performance to 405B model. So there might not be a need for the (future) work using a very large model for generating synthetic data?
> > >
> > > - Another related question is this, generating synthetic data using a 70B model might be costly as well. I'm wondering given the same budget (e.g., inference compute), how the accuracy of an 8B model trained on 8B model synthetic data compared with an 8B model trained on 70B model synthetic data? For the former, you will generate a lot more data samples while for the latter you will have more quality samples. Understanding this might further strengthen the contribution of the work. This will also help people choose their teacher model when doing synthetic data generation given a certain budget.

---

> ### Author Response · Authors · 2024-11-24
>
> Thanks! We're glad to have answered your questions.
>
> > From the new results, It seems that 70B model achieves comparable performance to 405B model. So there might not be a need for the (future) work using a very large model for generating synthetic data?
>
> Yes, from the results we presented this is indeed a valid conclusion.
> However, we would caution that these results are based on a small-scale experiment limited to the Solution-Augmentation portion of the data. From our experience, synthesizing high-quality questions is much harder than synthesizing solutions. In our experience so far, small models don't do a good job of synthesizing questions.
>
> > Another related question is this, generating synthetic data using a 70B model might be costly as well. I'm wondering given the same budget (e.g., inference compute), how the accuracy of an 8B model trained on 8B model synthetic data compared with an 8B model trained on 70B model synthetic data? For the former, you will generate a lot more data samples while for the latter you will have more quality samples. Understanding this might further strengthen the contribution of the work. This will also help people choose their teacher model when doing synthetic data generation given a certain budget.
>
> Based on [the inference cost](https://www.together.ai/pricing), there's a 5x cost difference between the 70B and 8B Llama models.
> We have previously done experiments at the scale of 64K dataset size synthesized by the 405B model on which the fine-tuned model scores around 41.5. Assuming the 70B synthesized data roughly gives the same performance and that reducing the dataset by a further 2.5x (to match the inference cost) should only worsen the performance, we can safely upper bound the result for the model fine-tuned using 70B synthesized data at 41.5. The performance for the 8B synthesized model is 42.96 with 128K data size, as reported above. Thus, the 8B model should be preferred for small-scale experiments with budget constraints.
>
> We still argue for using stronger models for data synthesis because we believe synthetic data generation is a one-time expense and a small portion of the overall compute budget of training LLMs.
>
> Hope this answers your question!

---

### Official Review · Reviewer_7gQ8 · 2024-11-04

**Soundness:** 3
**Presentation:** 4
**Contribution:** 3
**Rating:** 6
**Confidence:** 4

**Summary:**

This paper introduces OpenMathInstruct-2, a large open-source dataset for improving mathematical reasoning in language models. The authors propose a more concise Chain-of-Thought (CoT) format, which enhances performance compared to existing methods. Through extensive experiments, they demonstrate that using a strong teacher model for data generation is more effective than on-policy data from weaker models. The study also highlights the robustness of models to low-quality data and emphasizes the importance of question diversity in fine-tuning. Overall, the paper contributes to open-source resources and offers valuable insights into optimizing mathematical reasoning in language models.

**Strengths:**

1. Open-Source Dataset: This paper released the code, the finetuned models, and the OpenMathInstruct-2 dataset under a commercially permissive license.
2. Extensive Experiments: Not only main experiments, authors did a lot ablation study about: 1. choice of data generation model, 2. robustness of SFT, 3. impact of question diversity. The results are interesting and solid.

**Weaknesses:**

This is a solid work; I don't think there are any obvious weaknesses.

**Questions:**

1. What do you think about a self-improving framework, such as using an 8B Llama model to generate synthetic data and fine-tune it, then repeating the process for 2-3 rounds?
2. What do you think is the reason for the inadequate performance of on-policy data?

---

> ### Author Response · Authors · 2024-11-21
> **Response to Reviewer 7gQ8**
>
> We thank the reviewer for their kind words and respond to their queries below.
>
> > What do you think about a self-improving framework, such as using an 8B Llama model to generate synthetic data and fine-tune it, then repeating the process for 2-3 rounds?
>
> We believe that with appropriate data selection, the model can significantly improve performance over Llama3.1-8B’s performance with a self-improving framework.  It would be interesting to investigate when in this iterative process does the model performance start hitting the ceiling.
>
> > What do you think is the reason for the inadequate performance of on-policy data?
>
> We believe the subpar performance of finetuning Llama3.1-8B on its own synthesized data has to do with the quality of solutions. Table 6 in the Appendix shows that both LLM-as-a-Judge and the Nemotron reward model identify more low-quality solutions in Llama3.1-8B synthesized solutions in comparison to Llama3.1-405B-Instruct synthesized ones.

---

> > ### Comment · Reviewer_7gQ8 · 2024-11-25
> >
> > Thanks for your response.

---

### Official Review · Reviewer_Wcne · 2024-11-04

**Soundness:** 4
**Presentation:** 4
**Contribution:** 4
**Rating:** 8
**Confidence:** 4

**Summary:**

This paper introduces OpenMathInstruct-2, an SFT recipe for improving the math reasoning of language models. With extensive engineering efforts and experiments, the authors show the importance of solution formats, the need of a strong teacher, question diversity, and the robustness to imperfect solutions. Putting everything together, the authors show that OpenMathInstruct-2 improves Llama 3.1 and performs strongly compared to other open-source models.

**Strengths:**

I think this paper is in general a high-signal paper. Specifically:

- **Well-written with high-density of details**: this paper is well-written. The flow is a naturally interleaved summary, detail, and explanations. Many important details (size of model, size of dataset, specific accuracy numbers .etc) are packed in the writing with clear explanations of the motivation.
- **Well-execution**: the experiments in this paper is clear. I particularly appreciate the effort and discussion about decontamination, which essentially strengthens this paper quite a bit.
- **Sufficient signal from experiments**: the model developed in this paper is not top 1 in the leaderboard (but understandable because Qwen uses more resources). But the signal sent by the experiments (tolerate to imperfect annotation, importance of question diversity .etc) is clear.

**Weaknesses:**

There are two conclusions draw by the authors that I am not so sure:

- **Format of the data**. In general I agree that excessively verbose CoT may not be good for either performance or user preference, but it is not very clear how exactly the CoT proposed in this work contribute to the improved accuracy of reasoning. The higher numbers reported by the author is a good observation but not the mechanism behind. In addition, existing successful models like O1, in many cases, also offer verbose CoT, yet its performance is remarkable. In general, I tend to believe the level of detail that should be included in the CoT may still need further investigation, and I do see mixed signals favor or not favor verbose CoT.
- **Limited domain**: the authors still use GSM8K and MATH as the seed data — they do generate new questions but the seed is still GSM8K and MATH. The author test the performance on more challenging tasks (AMC and AIME), which is a nice practice. But I do feel that the field is moving forward to wider domains (not just math but also other STEM) and higher level (not just high-school but college). For this work I think it is sufficient to just use GSM8K and MATH, but I would suggest the authors to move forward in their future endeavors — only focusing on these two datasets would limit the authors from making higher impact.

**Questions:**

In Table 4, the authors compare with Qwen 2.5, Mathtral, NuminaMath and DS-Coder. I understand these are strong open-source baselines so comparing with them is sufficient to see the signal, but I would still encourage the authors include more OSS/ proprietary  models for further comparison (even other models are much weaker or stronger).

---

> ### Author Response · Authors · 2024-11-21
> **Response to Reviewer Wcne**
>
> We thank the reviewer for their kind words and detailed assessment.
>
> > In general I agree that excessively verbose CoT may not be good for either performance or user preference, but it is not very clear how exactly the CoT proposed in this work contribute to the improved accuracy of reasoning. The higher numbers reported by the author is a good observation but not the mechanism behind. In addition, existing successful models like O1, in many cases, also offer verbose CoT, yet its performance is remarkable. In general, I tend to believe the level of detail that should be included in the CoT may still need further investigation, and I do see mixed signals favor or not favor verbose CoT.
>
> We agree with the reviewer that verbosity has had mixed effects on performance, especially with the release of o1. However, there are two important distinctions in our setup in comparison to o1. Firstly, our models are not trained for self-correction which means the solutions tend to be shorter than ones in which models can recover from errors. We see such self-recovery patterns in Qwen-generated solutions as well. Secondly, the o1 model is presumably much bigger and has better long-context capabilities than the Llama3.1-8B model for which we present this ablation. So it is possible that bigger models with better long-context capabilities can perform at par or better with more verbose CoTs.
>
> In our setup, we also found that more verbose data might include some erroneous information, as lengthy CoTs can be a result of flawed reasoning. But overall, we agree that our findings might be limited to certain setups.
>
>
> > Limited domain: the authors still use GSM8K and MATH as the seed data — they do generate new questions but the seed is still GSM8K and MATH. The author test the performance on more challenging tasks (AMC and AIME), which is a nice practice. But I do feel that the field is moving forward to wider domains (not just math but also other STEM) and higher level (not just high-school but college). For this work I think it is sufficient to just use GSM8K and MATH, but I would suggest the authors to move forward in their future endeavors — only focusing on these two datasets would limit the authors from making higher impact.
>
> We thank the reviewer for this suggestion. Given the performance saturation on GSM8K and MATH, we’ve indeed shifted our focus to much harder benchmarks. But we want to highlight that the performance saturation on MATH is pretty recent, and the best open-weight models were still in the low 70s till a few months back.
>
> > In Table 4, the authors compare with Qwen 2.5, Mathtral, NuminaMath and DS-Cder. I understand these are strong open-source baselines so comparing with them is sufficient to see the signal, but I would still encourage the authors include more OSS/ proprietary models for further comparison (even other models are much weaker or stronger).
>
> We opted for a limited selection of models in Table 4 since we wanted to limit the focus to high-performant, open-source/weight models. We can add an extensive version of this table, as the reviewer suggests, in the Appendix.

---

### Comment · Area_Chair_a2Tq · 2024-11-27

Dear reviewers,

Thank you for your efforts reviewing this paper. If you haven't, can you please check the authors' responses and see if your concerns have been addressed? Please acknowledge you have read their responses. Thank you!

---

### Meta-Review · Area_Chair_a2Tq · 2024-12-21

**Metareview:**

Summary:

This paper aims to create a high-quality finetuning dataset for math reasoning, and conduct careful ablation experiments on data synthesis using the recently released Llama 3.1 family of models. The experiments show the importance of solution formats, the need of a strong teacher, question diversity, and the robustness to imperfect solutions. This paper creates the OpenMathInstruct-2 dataset which consists of 14M question-solution pairs (≈ 600K unique questions), which is nearly eight times larger than the previous largest open-source math reasoning dataset. Finetuning the Llama-3.1-8B-Base using OpenMathInstruct-2 outperforms Llama3.1-8B-Instruct on MATH by an absolute 15.9% (51.9% → 67.8%). The paper releases the code, the finetuned models, and the OpenMathInstruct-2 dataset under a commercially permissive license.

Strengths:

1. Well-written with great details and easy to follow
2. Experiments are well-executed.
3. Valuable findings that advance the understanding of LLM synthetic data generation in math reasoning.

Weaknesses:

There seems to be no major weakness after the discussion period. Most comments and questions raised by the reviewers have been addressed.

**Additional Comments On Reviewer Discussion:**

1. Reviewer Wcne raises a concern about the conclusion related to verbose CoT and calls for further investigation on how the level of details included in CoT affects the performance. The authors provides further clarification during the discussion period.

2. Reviewer vtSB raises concerns about the quality of generated questions and teacher model selection. These concerns seem to have been addressed (at least partially).

3. Reviewer uMxv raises concerns about the novelty of the process of dataset generation and some findings, but increases the score after the authors provide further clarifications.

---

### Decision · Program_Chairs · 2025-01-22

Accept (Poster)